## PERSPECTIVES

### Selective stimulation with intraneural electrodes for bionic limb prostheses can contribute to shed light on human touch sensorimotor integration

**Calogero Maria Oddo** 

*The BioRobotics Institute and Department of Excellence in Robotics and AI, Scuola Superiore Sant'Anna, Pisa, Italy*

Email: calogero.oddo@santannapisa.it

Edited by: Richard Carson & Vaughan Macefield

Linked articles: This Perspectives article highlights an article by Ranieri *et al*. To read this paper, visit https://doi.org/10.1113/JP282259.

The peer review history is available in the Supporting Information section of this article (https://doi.org/10.1113/JP282734#support-information-section).

Stimulus delivery is a major challenge in the investigation of sensory physiology and physiopathology, with translational implications while targeting bionic substitution of impaired sensorimotor functions by means of neuroprostheses. The complexity represented by stimulus administration is generally valid for all senses, and particularly for touch among physical senses because of the distributed positioning of receptors throughout the large area of the whole skin and the intrinsically interactive characteristics of somatosensation.

In touch studies, the experimental protocols typically involve the delivery of mechanothermal or electrical stimuli and the collection and analysis of stimulus-induced responses, either at perceptual level by means of psychophysical methods or by investigating peripheral or central bioelectronic activity in humans or animals via techniques such as microneurography and microstimulation, patch clamp, electrode arrays, functional magnetic resonance imaging, or electroencephalography.

When the stimulus is delivered mechanically directly to the epidermis of intact subjects, such as with vibrotactile probes, extended textures or shapes, the spatiotemporal interaction with the skin should be carefully controlled over multiple sessions, for example by using custom biorobotic tools. Such automatic mechatronic platforms should be developed to allow gathering of repeatable responses of the somatosensory pathways in order to enable modelling attempts based on brain imaging or electrophysiological recordings at higher stages with respect to the cutaneous haptic stimulation site.

Another suitable approach involves accessing directly the nervous system in order to release electrical pulses to the afferent pathways. The electrical delivery of the stimuli is extremely challenging because of the complexity of guaranteeing mechanical, electrical and biological stability of the electrode-nerve interface, the high number of stimulation sites, and selectivity, both spatially and temporally. Among the possible methods for releasing electrical patterns to the nervous system, microstimulation with needle electrodes is a technique used by trained neurophysiologists for eliciting action potentials with excellent axonal selectivity within a microneurographic session (Vallbo, 2018). While being minimally invasive, and thus applicable in trials involving either intact or amputee humans, microstimulation has the inherent constraint of single-site electrode placement and stability limited to short term protocols in acute subjects. However, progress with neurotechnologies and neurosurgical techniques has enabled over the past decade a series of studies that are demonstrating the feasibility of chronically implanting electrodes in the peripheral nerves of human subjects for neuroprosthetic applications, such as perineural cuffs or intraneural filament electrodes that are being applied in pilot clinical trials with limb amputees to partially restore sensorimotor functions (Farina *et al*. 2021).

The study coordinated by Di Pino, published in this issue of The *Journal of Physiology*, builds on these recent technological and medical advancements (Ranieri *et al*. 2022). The paper reports a case study involving a 40-year-old transradial amputee surgically implanted with perineural cuffs and intraneural double-sided filament electrodes (ds-FILE) in the median and ulnar nerves. The implemented experimental protocol consisted of administering electrical pulses via the implanted perineural and intraneural interfaces, as well as by means of control transcutaneous stimulation, to evaluate the inhibitory effect of somatosensory stimulation on contralateral motor cortex excitability. As initial finding, the specificity of the delivered stimulation was assessed by recording a very large dataset of scalp somatosensory evoked potentials (SEPs); the resulting data challenge current thinking with respect to SEP amplitude in response to nerve stimulation, which in the present study with intraneural interfaces (Ranieri *et al*. 2022) was much lower when compared to previous reports involving less selective electrodes. The core of the experimental protocol, enabled by the chronically implanted electrodes, was very elegant and included carefully designed controls: transcranial magnetic stimulation (TMS) was administered on the motor cortex to induce motor evoked potentials (MEPs) revealed via electromyographic (EMG) recordings in contralateral muscles, whereas somatosensory electrical stimuli were delivered via all available interfaces (intraneural, perineural, and transcutaneous) with variable stimulation properties such as active electrode location and latency with respect to TMS. The main finding is the demonstration of short-latency afferent inhibition (Tokimura *et al*. 2000) on motor cortex induced by intraneural stimulation only under specific combinations of active sites and timing. This finding was claimed to be the first reported case of selective low-latency inhibition of motor cortical output by means of intraneural interfaces, which presumably acts via thalamo-cortical pathways for sensorimotor integration. Though the experimental protocol involved a single subject and therefore the findings should be replicated in future trials to assess their generalizability, methodologically the study reported by Ranieri and colleagues consolidates evidence of the prospective role of neuroprosthetic research not only for its clinical potential but also as a discovery engine to investigate and model the sense of touch (Oddo *et al*. 2016). Although the long-term stability of neural implants is still an open issue to be further evaluated after the recent first-in-human trials (Farina *et al*. 2021), the improved

The Journal of Physiology

selectivity, number of stimulation sites and long-term stability guaranteed by present neurotechnologies offer a chronic access to the nervous system. This possibility can enable unprecedented experimental protocols complementing other electrophysiological and imaging techniques to advance the fundamental knowledge of physiology and physiopathology of the human sensorimotor system.

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

## Additional information

### Competing interests

No competing interests declared.

### Author contributions

Sole author.

### Funding

The ideas discussed in this perspective paper were supported in part by the Tuscany Region through the TUscany NEtwork for BioElectronic Approaches in Medicine: AI based predictive algorithms for fine-tuning of electroceutics treatments in neurological, cardiovascular and endocrinological diseases (TUNE-BEAM, n. H14I20000300002) and by the European Commission through the NEuro-controlled BIdirectional Artificial upper limb and hand prosthesiS project (NEBIAS, n. 611687).

### Acknowledgement

Open Access Funding provided by Scuola Superiore Sant'Anna within the CRUI-CARE Agreement.

### Keywords

compound muscle action potentials, electroencephalogram, intraneural stimulation, perineural stimulation, short-latency afferent inhibition, somatosensory evoked potentials, transcranial magnetic stimulation, transcutaneous stimulation

## Supporting information

Additional supporting information can be found online in the Supporting Information section at the end of the HTML view of the article. Supporting information files available:

**Peer Review History**

