## [Peer Review History · The Journal of Physiology]

Selective stimulation with intraneural electrodes for bionic limb prostheses can contribute to shed light on human touch sensorimotor integration

Calogero Maria Oddo
DOI: 10.1113/JP282734

Corresponding author(s): Calogero Oddo (calogero.oddo@santannapisa.it)

The following individual(s) involved in review of this submission have agreed to reveal their identity: Federico Ranieri (Referee #1)

Review Timeline:

Submission Date:	25-Dec-2021
Editorial Decision:	10-Jan-2022
Revision Received:	10-Jan-2022
Editorial Decision:	11-Jan-2022
Revision Received:	11-Jan-2022
Accepted:	13-Jan-2022

Senior Editor: Richard Carson

Reviewing Editor: Vaughan Macefield

Transaction Report:

Dear Dr Oddo,

Re: JP-P-2021-282734 "Selective stimulation with intraneural electrodes for bionic limb prostheses can contribute to shed light on human touch sensorimotor integration" by Calogero Maria Oddo

Thank you for submitting your invited Perspectives article to The Journal of Physiology. It has been assessed by a Reviewing Editor and the author of the focus paper.

Minor alterations have been requested.

The reports are copied at the end of this email. Please address all of the points and incorporate all requested revisions.

NEW POLICY: In order to improve the transparency of its peer review process The Journal of Physiology publishes online as supporting information the peer review history of all articles accepted for publication. Readers will have access to decision letters, including all Editors' comments and referee reports, for each version of the manuscript and any author responses to peer review comments. Referees can decide whether or not they wish to be named on the peer review history document.

I hope you will find the comments helpful and have no difficulty in revising your article within 7 days.

To submit the revised version use the links in Author Tasks Link Not Available.

Please ensure that the article is a Word File with no more than 5 references, including the focus paper.

Thank you for your contribution to the Journal.

Yours sincerely,

Richard Carson
Senior Editor
The Journal of Physiology

EDITOR COMMENTS

Reviewing Editor:

Thank you for agreeing to write this Perspectives article. The corresponding author of the manuscript to which your article refers is pleased with your interpretation of the manuscript, but requests some minor amendments before we can accept your article. I trust these can be attended to promptly.

REFEREE COMMENTS:

Referee #1:

This perspective article properly interprets the contents of the focus paper, and it provides a clear overview on its translational implications.

I only suggest a few minor revisions.

- "SEP amplitude" (or "size", or "magnitude") should preferably be used instead of "SEP intensity", also to avoid confusion with stimulation intensity.

- The following sentence can be made more precise with respect to discussed findings: "The main finding is the demonstration of short-latency afferent inhibition (Tokimura et al., 2000) on motor cortex induced by intraneural stimulation

only under specific combinations of active electrodes and latency, while stimulation with the transcutaneous interface was less specific and the inhibition was systematically observed for all tested muscles". Specifically: a) I would not compare with transcutaneous stimulation here as it is non-selective, and different muscle targets (distal and proximal) have not been tested with intraneural stimulation; b) the term "sites" could be used instead of "electrodes" to avoid confusion with the term "electrode" previously used in the text to describe the entire interface; c) I would replace the term "latency" with "timing", considering that "latency" typically indicates an event following a previous one (here we have the nerve stimulus preceding TMS). Therefore, a possible rephrasing is: "The main finding is... only under specific combinations of stimulation sites and timing."

- In the sentence "This finding was claimed to be the first reported case of selective low-latency inhibition of motor cortical output...", it should be specified "by intraneural interfaces", to take into account previous reports on SAI obtained by selective cutaneous stimulation (e.g., Tamburin et al. 2005; Pilurzi et al. 2020).

- It could be considered mentioning that robustness (i.e., long-term stability) of intraneural interfaces still represents an issue for chronic interfacing with the nervous system.

Neuro-Robotic Touch Laboratory

Prof. Dr. Calogero M Oddo, Head
calogero.oddo@santannapisa.it
The BioRobotics Institute
Department of Excellence in Robotics & A.I.
Scuola Superiore Sant'Anna
Viale R. Piaggio 34, Pontedera (Pisa), Italy

Pisa, January 10, 2022

Dear Editor and Referee,

Let me thank you again for the invitation to submit a perspective paper stemming from the study entitled "Sensorimotor integration within the primary motor cortex by selective nerve fascicle stimulation", and for the positive evaluation of the manuscript that I submitted to the attention of Journal of Physiology.

Enclosed, please find the revised manuscript, which integrates the constructive minor revision requests received.

In the submission, please find both the clean version of the document and the revision with track-changes highlighting the amendments applied.

Please do not hesitate to contact me in case of any action that may be needed on my side.

Thanks again for the invitation and for the positive evaluation, and best regards,

Calogero Maria Oddo

Dear Dr Oddo,

Re: JP-P-2022-282734R1 "Selective stimulation with intraneural electrodes for bionic limb prostheses can contribute to shed light on human touch sensorimotor integration" by Calogero Maria Oddo

Thank you for submitting your invited revised Perspectives article to The Journal of Physiology. It has been assessed by a Reviewing Editor and the author of the focus paper.

Minor alterations have been requested.

The reports are copied at the end of this email. Please address all of the points and incorporate all requested revisions.

NEW POLICY: In order to improve the transparency of its peer review process The Journal of Physiology publishes online as supporting information the peer review history of all articles accepted for publication. Readers will have access to decision letters, including all Editors' comments and referee reports, for each version of the manuscript and any author responses to peer review comments. Referees can decide whether or not they wish to be named on the peer review history document.

I hope you will find the comments helpful and have no difficulty in revising your article within 7 days.

To submit the revised version use the links in Author Tasks Link Not Available.

Please ensure that the article is a Word File with no more than 5 references, including the focus paper.

Thank you for your contribution to the Journal.

Yours sincerely,

Richard Carson
Senior Editor
The Journal of Physiology

EDITOR COMMENTS

Reviewing Editor:

Thank you for attending to the Reviewer's minor comments.

Please correct the typographical error in the following statement: "Although the long-term stability of neural implants is still and open issue..." [replace "and" with 'an']

In addition, I have uncovered some further grammatical errors that require your attention:

Replace "...grounds on these recent technological..." with "...builds on these recent technological..."

Replace "...was much lower if compared to..." with "...was much lower when compared to..."

Replace "...which could presumably be associated to thalamo-cortical..." with "...which presumably acts via thalamo-cortical..."

Neuro-Robotic Touch Laboratory

Prof. Dr. Calogero M Oddo, Head
calogero.oddo@santannapisa.it
The BioRobotics Institute
Department of Excellence in Robotics & A.I.
Scuola Superiore Sant'Anna
Viale R. Piaggio 34, Pontedera (Pisa), Italy

Pisa, January 11, 2022

Dear Editor and Referee,

Let me thank you again for the invitation to submit a perspective paper stemming from the study entitled “Sensorimotor integration within the primary motor cortex by selective nerve fascicle stimulation”, and for the positive evaluation of the manuscript that I submitted to the attention of Journal of Physiology.

Enclosed, please find the revised manuscript, which integrates the editorial changes proposed. Many thanks for the careful assessment of the manuscript, to enhance its quality.

In the submission, please find both the clean version of the document and the revision with track-changes highlighting the amendments applied.

Please do not hesitate to contact me in case of any action that may be needed on my side.

Thanks again for the invitation and for the positive evaluation, and best regards,

Calogero Maria Oddo

Dear Dr Oddo,

Re: JP-P-2022-282734R2 "Selective stimulation with intraneural electrodes for bionic limb prostheses can contribute to shed light on human touch sensorimotor integration" by Calogero Maria Oddo

I am pleased to tell you that your invited Perspective article has been accepted for publication in The Journal of Physiology.

NEW POLICY: In order to improve the transparency of its peer review process The Journal of Physiology publishes online as supporting information the peer review history of all articles accepted for publication. Readers will have access to decision letters, including all Editors' comments and referee reports, for each version of the manuscript and any author responses to peer review comments. Referees can decide whether or not they wish to be named on the peer review history document.

The last Word version of the paper submitted will be used by the Production Editors to prepare your proof. When this is ready you will receive an email containing a link to Wiley's Online Proofing System. The proof should be checked and corrected as quickly as possible.

All queries at proof stage should be sent to tjp@wiley.com

Thank you very much for your contribution to The Journal of Physiology.

Yours sincerely,

Richard Carson
Senior Editor
The Journal of Physiology

Reviewing Editor Comments:

Thank you for attending to these minor concerns. I am now happy to recommend acceptance of your manuscript.